# Mitochondrial membrane potential acts as a retrograde signal to regulate cell cycle progression

Choco Michael Gorospe[1], Gustavo Carvalho[1], Alicia Herrera Curbelo[1], Lisa Marchhart[1], Isabela C Mendes[1], Katarzyna Niedźwiecka[1,2], Paulina H Wanrooij[1]

**Mitochondria are central to numerous metabolic pathways whereby mitochondrial dysfunction has a profound impact and can manifest in disease. The consequences of mitochondrial dysfunction can be ameliorated by adaptive responses that rely on crosstalk from the mitochondria to the rest of the cell. Such mito-cellular signalling slows cell cycle progression in mitochondrial DNA–deficient ($\rho^0$) *Saccharomyces cerevisiae* cells, but the initial trigger of the response has not been thoroughly studied. Here, we show that decreased mitochondrial membrane potential ($\Delta\Psi m$) acts as the initial signal of mitochondrial stress that delays G1-to-S phase transition in both $\rho^0$ and control cells containing mtDNA. Accordingly, experimentally increasing $\Delta\Psi m$ was sufficient to restore timely cell cycle progression in $\rho^0$ cells. In contrast, cellular levels of oxidative stress did not correlate with the G1-to-S delay. Restored G1-to-S transition in $\rho^0$ cells with a recovered $\Delta\Psi m$ is likely attributable to larger cell size, whereas the timing of G1/S transcription remained delayed. The identification of $\Delta\Psi m$ as a regulator of cell cycle progression may have implications for disease states involving mitochondrial dysfunction.**

## Introduction

Mitochondria use the electron transport chain (ETC) complexes to convert energy gained from the oxidation of nutrients into an electrochemical gradient across the inner mitochondrial membrane that is then used to drive ATP synthesis through oxidative phosphorylation (OXPHOS). In addition to producing most of the cell's ATP, however, mitochondria carry out a diverse array of vital cellular functions including synthesis of Fe-S clusters, amino acid and nucleotide biosynthesis, the production of reactive oxygen species (ROS), and apoptosis. It follows that mitochondrial function is required for cell survival even in a facultative anaerobe like the budding yeast *Saccharomyces cerevisiae* that can survive without

mitochondrial ATP production. In line with the manifold functions of the organelle, mitochondrial dysfunction is associated with numerous diseases including neurodegenerative disorders, metabolic syndrome, cancer, and ageing (1, 2). Rather than manifesting merely as an energy defect, mitochondrial dysfunction can contribute to cellular dysfunction and disease aetiology through diverse mechanisms involving, for example, increased levels of ROS that can damage cellular constituents, changes in nuclear epigenetic marks or gene expression patterns, and even instability of the nuclear genome (1, 3, 4, 5).

To avoid or ameliorate the dire consequences of mitochondrial dysfunction, a complex communication network mediates signals of mitochondrial status to other parts of the cell including the nucleus, lysosomes, peroxisomes, and the endoplasmic reticulum (1, 2). This mito-cellular signalling can involve signals ranging from key metabolites to misfolded proteins or ROS, and acts to restore cellular homeostasis, facilitate adaptation to the altered mitochondrial status, or eliminate dysfunctional mitochondria via mitophagy (6).

Mitochondria contain their own genome, mtDNA, that encodes key subunits of the ETC complexes, as well as the $F_o$ component of the ATP synthase, making this small genome essential for cellular respiration and OXPHOS. It follows that cells lacking mtDNA (termed $\rho^0$ cells) cannot maintain the mitochondrial membrane potential ($\Delta\Psi m$) by proton pumping; instead, they maintain a limited $\Delta\Psi m$ by a mechanism involving the hydrolysis of ATP by the "reverse" action of the $F_1$ component of the ATP synthase in consort with the electrogenic exchange of $ATP^{4-}$ for $ADP^{3-}$ by the adenine nucleotide translocator (7, 8, 9). Accordingly, certain hypermorphic mutations in subunits of the $F_1$-ATPase result in a higher membrane potential in $\rho^0$ cells (10).

In addition to preventing cellular respiration and decreasing $\Delta\Psi m$, the acute depletion of mtDNA after a brief ethidium bromide treatment promotes nuclear DNA instability and defective cell cycle progression that manifests as an accumulation of cells in the G1 phase (5). Moreover, although the nuclear DNA instability in mtDNA-depleted cells is driven by defective Fe-S cluster protein assembly, the cell cycle defect was found to be independent of

[1]Department of Medical Biochemistry and Biophysics, Umeå University, Umeå, Sweden [2]Institute of Biochemistry and Biophysics, Polish Academy of Sciences, Warsaw, Poland

Correspondence: paulina.wanrooij@umu.se

Fe-S metabolism ([5](5)). Subsequent work elaborated that a cell cycle phenotype of mtDNA-depleted cells is seen even in cells that are adapted to their $\rho^0$ status, and is attributable to delayed transition from the G1 to the S phase of the cell cycle through a mechanism proposed to involve the Rad53 checkpoint kinase ([11](11)). However, despite the undisputed involvement of mitochondrial dysfunction in human disease, the molecular-level events that link mtDNA loss to the cell cycle machinery, and particularly the mitochondria-proximal signal that initiates this form of mitocellular communication, have not been thoroughly addressed. Here, we sought to identify the mechanism that triggers the delayed G1-to-S progression in $\rho^0$ cells. A combination of genetic and pharmaceutical interventions uncovered decreased mitochondrial membrane potential as a regulator of cell cycle progression not only in $\rho^0$ cells, but also in $\rho^+$ cells. Accordingly, the cell cycle defect in $\rho^0$ cells was rescued by increasing $\Delta\Psi m$, confirming that adequate $\Delta\Psi m$ rather than full mitochondrial function per se is required for normal cell cycle progression. In contrast, neither inhibition of mitochondrial ATP synthesis nor altered levels of oxidative stress could directly account for

the G1-to-S transition delay of $\rho^0$ cells. These findings corroborate that mitochondrial function, via mitochondrial membrane potential, acts as a regulator of cell cycle progression, and call for a better understanding of the mechanisms through which dysfunctional mitochondria impact the cell cycle machinery.

# Results

## Decreased mitochondrial membrane potential delays G1-to-S phase progression

To study the effects of mtDNA loss, an mtDNA-devoid ($\rho^0$) variant of the mtDNA-containing ($\rho^+$) WT strain was made by standard treatment with ethidium bromide ([12](12)), and confirmed by quantitative real-time PCR. In accordance with previous reports, $\rho^0$ cells grew slower than their $\rho^+$ counterparts in a dextrose-containing rich medium, and exhibited an altered cell cycle profile ([Figs 1A](Fig1A) and [S1A](FigS1A)) ([11](11), [13](13)). Specifically, the percentage of G1 cells in an early logarithmic phase–unsynchronized culture of $\rho^0$ cells was twice as high

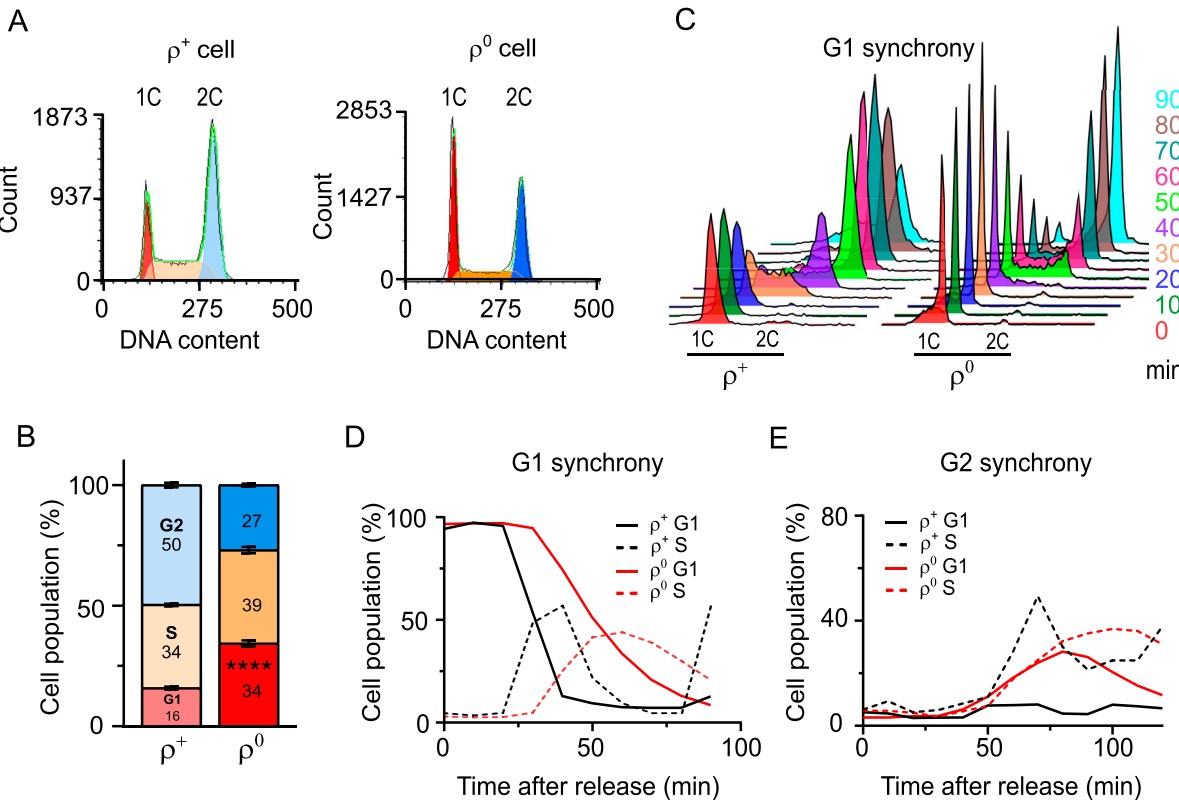

**Figure 1.  Loss of mtDNA induces a delay in transition from the G1 to the S phase of the cell cycle.**
**(A)** Representative DNA histogram of unsynchronized WT (AC402) $\rho^+$ and $\rho^0$ cells grown to an early logarithmic phase in YPDA. *1C* and *2C* indicate populations with single and double chromosome contents, corresponding to cells in G1 and G2, respectively. The cell cycle profile was analysed using the multicycle model in FCS Express to determine the percentage of cells in G1 (red), S (orange), and G2 (blue) phases. **(B)** Quantification of the percentage of cells in the G1, S, or G2 phase. Values represent the average of at least four independent experiments including the one in [Fig 1A](Fig1A), and error bars indicate SD. The two-tailed *t* test was performed to determine statistical significance between the G1 populations in WT $\rho^+$ and $\rho^0$ cultures. ****$P < 0.0001$. **(C)** DNA histogram of WT (AC402) $\rho^+$ and $\rho^0$ cells released from G1 synchrony achieved by treatment with 10 μg/ml α-factor. Cells were sampled every 10 min after release. The experiment was repeated at least three times; a representative experiment is shown. **(D)** Quantification of the percentage of G1 phase (*solid lines*) and S phase (*dashed lines*) cells after release from G1 synchrony in the experiment shown in [Fig 1D](Fig1D). **(E)** Quantification of the percentage of G1 phase (*solid lines*) and S phase (*dashed lines*) cells after release of WT $\rho^+$ and $\rho^0$ cells from G2 synchrony achieved by nocodazole treatment. The DNA histograms are shown in [Fig S1D](FigS1D). Values represent data from a single experiment. See also [Fig S1](FigS1).

as in $\rho^+$ cells grown under identical conditions; this difference remained constant for at least 2 h during exponential growth and was also observed in cells that had lost their mtDNA because of deficiency of known mtDNA maintenance factors such as Rim1, Mip1, and Mgm101 (Figs 1A and B and S1B and C). The increased frequency of G1 cells in the unsynchronized $\rho^0$ culture was attributable to a slower G1-to-S transition rather than a faster progression through other cell cycle stages because α-factor–treated $\rho^0$ cells released from G1 synchrony transitioned into the S phase later than their $\rho^+$ counterparts (Fig 1C and D; compare timepoints when % of G1 cells starts to decrease or % of S cells peaks in panel D), corroborating previous findings (11). A G1-to-S transition delay of the $\rho^0$ variant was observed even after release from G2 synchrony achieved by nocodazole treatment: although $\rho^+$ and $\rho^0$ cells showed comparable timing of G2 exit (Fig S1D and E), $\rho^0$ cells accumulated in G1 and displayed delayed entry into the S phase when compared to $\rho^+$ cells (Fig 1E). Using our mtDNA-deficient strain that exhibited the previously reported delay in G1/S transition (5, 11, 14), we proceeded to seek the identity of the proximal signal that triggers the cell cycle phenotype in mtDNA-deficient cells.

Yeast mtDNA encodes proteins that are essential for the ETC and OXPHOS. The impairment of these processes in $\rho^0$ cells results in compromised mitochondrial ATP production, decreased ΔΨm, and increased generation of ROS (15, 16, 17), all of which are reasonable candidates for the initial signal that triggers the G1-to-S phase

delay in $\rho^0$ cells. In line with a previous report (11), treatment of $\rho^+$ cells with the ATP synthase inhibitor oligomycin did not cause accumulation of cells in the G1 phase (Figs 2A and S2A); the potency of the oligomycin was confirmed by its effect on mitochondrial membrane potential measured using the fluorescent cationic dye tetramethylrhodamine methyl ester perchlorate (TMRE) (Fig S2B, lower panel). As expected based on the lack of a functional $F_o$ subunit, $\rho^0$ cells were not impacted by oligomycin treatment (Fig S2B, upper panel). Decreased mitochondrial ATP synthesis was therefore excluded as a cause of the G1-to-S transition delay observed in $\rho^0$ cells.

We next tested the importance of the ΔΨm for normal cell cycle progression. To this end, we treated exponentially growing cells with BAM15, an uncoupler that efficiently dissipates the ΔΨm without depolarizing the plasma membrane (18). Interestingly, BAM15 treatment of $\rho^+$ cells caused a transient accumulation of cells in the G1 phase (Figs 2B and S2C). The observed accumulation of cells in the G1 phase was accompanied by a corresponding decrease in the percentage of cells in the S phase. Treatment with another uncoupler, CCCP, caused a similar transient increase in G1 phase cells as BAM15 (Fig S2D and E).

To further corroborate the connection between ΔΨm and G1-to-S progression, we synchronized $\rho^+$ cells in G1 with α-factor and released them into a medium containing BAM15. In analogy to the G1-to-S transition delay in $\rho^0$ cells, BAM15 treatment delayed the

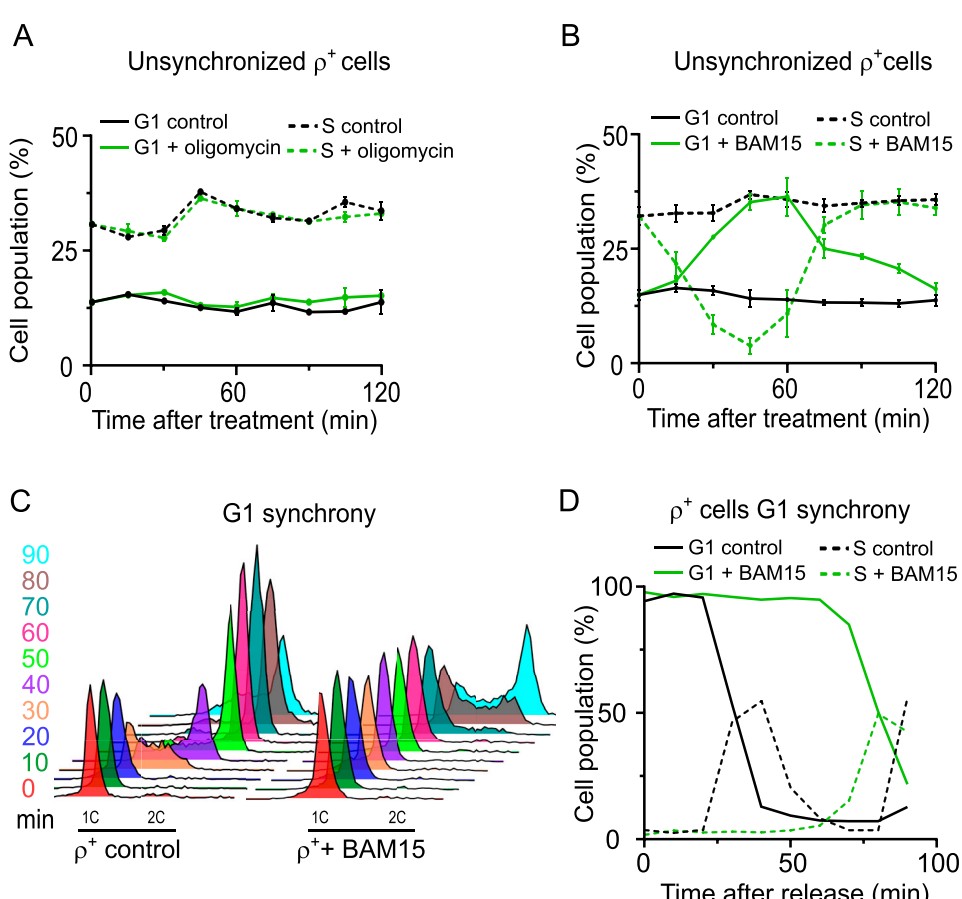

**Figure 2. Loss of ΔΨm, but not the inhibition of mitochondrial ATP synthesis, delays G1-to-S progression in WT $\rho^+$ cells.**
**(A)** Percentage of G1 phase (*solid lines*) and S phase (*dashed lines*) cells in WT $\rho^+$ in an early logarithmic phase left untreated or treated with 20 μM of oligomycin to inhibit mitochondrial ATP synthesis. Aliquots were harvested upon the addition of the drug (0 min) and every 15 min thereafter. Values represent the average of two independent experiments, and error bars indicate SD. Representative DNA histograms are shown in Fig S2A. **(B)** Percentage of G1 phase (*solid lines*) and S phase (*dashed lines*) cells in early logarithmic phase cultures of WT $\rho^+$ cells left untreated or treated with 30 μM of BAM15. Aliquots were harvested upon the addition of the drug (0 min) and every 15 min thereafter. Values represent the average of three independent experiments, and error bars indicate SD. Representative DNA histograms are shown in Fig S2C. **(C)** DNA histogram of WT $\rho^+$ cells synchronized in G1 with 10 μg/ml α-factor and released into media with or without 30 μM BAM15. Cells were sampled upon release (time 0) and every 10 min thereafter. **(C, D)** Quantification of G1 phase (*solid lines*) and S phase (*dashed lines*) cells in the experiment presented in panel (C). Values represent data from a single experiment. See also Fig S2.

progression of $\alpha$-factor–synchronized $\rho^+$ cells from the G1 into the S phase: although the percentage of G1 cells began to decline 20 min after release of $\rho^+$ cells into a normal medium, indicating transition from the G1 into the S phase, $\rho^+$ cells released into the BAM15-containing medium did not show signs of G1 exit until 70 min after release (Fig 2C and D). BAM15 treatment also delayed G1-to-S transition in $\rho^+$ cells released from G2 synchrony: although the presence of BAM15 did not appreciably affect the transition from the G2 to the G1 phase, $\rho^+$ cells released from G2 into the BAM15-containing medium showed an increased percentage of G1 cells and delayed S phase entry, consistent with a specific G1-to-S transition delay (Fig S2G–J).

The results of Fig 2 demonstrate that a $\Delta\Psi m$ collapse induced by uncoupler treatment causes cells to accumulate in the G1 phase by triggering a G1-to-S phase transition delay similar to the one observed in $\rho^0$ cells. These findings implicate $\Delta\Psi m$ in modulating cell cycle progression in $\rho^+$ cells.

## The cell cycle defect in $\rho^0$ cells is exacerbated by a further decrease in membrane potential

In contrast to uncoupler-treated cells where the $\Delta\Psi m$ is largely dissipated, $\rho^0$ cells have the ability to maintain a membrane potential that, although lower than in $\rho^+$ cells, is still sufficient to support various $\Delta\Psi m$-dependent processes such as the import of nuclear-encoded mitochondrial proteins (9, 19). Next, we examined whether the decreased $\Delta\Psi m$ could underlie the cell cycle phenotype of $\rho^0$ cells and act as the initial signal to trigger the G1-to-S transition delay. We measured the $\Delta\Psi m$ of WT $\rho^+$ and $\rho^0$ cells using TMRE, and corrected the signal for potential changes in strain mitochondrial mass as detected by nonyl acridine orange (NAO), a green-fluorescent dye that localizes to the mitochondria in a $\Delta\Psi m$-independent manner (20). Microscopy analysis confirmed the uptake of NAO, and thus its suitability as a readout of mitochondrial mass, into both $\rho^+$ and $\rho^0$ cells (Fig S3A). Uncoupler-treated samples analysed in parallel provided a measure of the background fluorescence. As expected, the $\Delta\Psi m$ of WT $\rho^0$ cells was significantly lower than that of WT $\rho^+$ cells, but clearly higher than the baseline value of uncoupler-treated cells (Figs 3A and S3B and C).

In additional support of a role for $\Delta\Psi m$ in regulating cell cycle progression, the accumulation of $\rho^0$ cells in the G1 phase was further exacerbated when $\Delta\Psi m$ was fully dissipated by BAM15 or CCCP treatment (Figs 3B and S2C, D, and F). The effect of BAM15 was more sustained in $\rho^0$ than in $\rho^+$ cells and was evident even 2 h after the addition of the compound (compare Figs 2B and 3B). Expectedly, time-course experiments after release from $\alpha$-factor– or nocodazole-induced synchrony revealed a more severe G1-to-S phase progression delay in $\rho^0$ cells in the presence of BAM15 (G1 exit starting 40 min and 80 min after release from $\alpha$-factor synchrony into the medium lacking and containing BAM15, respectively) (Figs 3C and D and S2G–J). These observations establish a quantitative correlation between $\Delta\Psi m$ and cell cycle progression, where the extent of the G1-to-S delay is governed by the severity of $\Delta\Psi m$ loss.

## Increasing $\Delta\Psi m$ rescues the cell cycle phenotype of $\rho^0$ cells

The metabolism of $\rho^0$ cells differs significantly from that of $\rho^+$ cells given that respiratory-deficient $\rho^0$ cells must compensate

for the loss of a subset of the citric acid cycle reactions, the products of which are central to many anabolic pathways (21). If the G1-to-S phase progression delay is indeed driven by decreased $\Delta\Psi m$ and not by other functional or metabolic differences between $\rho^+$ and $\rho^0$ cells, we hypothesized that it should be rescued by increasing $\Delta\Psi m$ in $\rho^0$ cells. In the absence of ETC activity, mtDNA-deficient cells maintain $\Delta\Psi m$ by an alternative mechanism that involves the hydrolysis of glycolytically produced ATP by the "reverse" action of the $F_1$ subunit of the mitochondrial ATP synthase and the electrogenic exchange of $ATP^{4-}$ for $ADP^{3-}$ over the inner mitochondrial membrane (7, 9). The *ATP1-111* mutation in the Atp1 subunit of the mitochondrial $F_1$-ATPase results in a hyperactive enzyme that generates a higher $\Delta\Psi m$ in $\rho^0$ cells than the one maintained in WT $\rho^0$ cells, and improves their growth (5, 19, 22). Accordingly, the $\Delta\Psi m$ of *ATP1-111* $\rho^0$ cells was comparable to that of WT $\rho^+$ cells, confirming the *ATP1-111* $\rho^0$ strain as a suitable model to test our hypothesis in Figs 4A and S3B and C.

We first analysed the cell cycle profile of unsynchronized WT and *ATP1-111* $\rho^+$ and $\rho^0$ cells. As expected based on the relatively high $\Delta\Psi m$ in *ATP1-111* $\rho^+$ cells, the strain showed a cell cycle profile that was indistinguishable from WT $\rho^+$ cells (Fig 4B and C). Moreover, analysis of $\alpha$-factor–synchronized cells revealed that the *ATP1-111* mutation was largely able to rescue the delay in G1-to-S transition observed in $\rho^0$ cells: although the percentage of G1 cells in cultures of WT and *cox4$\Delta$* $\rho^0$ cells was still above 90% 30 min after release from synchrony, only 67% of *ATP1-111* $\rho^0$ cells remained in the G1 phase (Figs 4D and S3D). Finally, the growth of *ATP1-111* $\rho^0$, as judged by colony size or cell density measurements in liquid media, was partially restored relative to WT $\rho^0$ cells (Figs 4E and S3E), as previously reported by others (5, 22).

The earlier experiments carried out on WT cells revealed a more sustained effect of BAM15 on the cell cycle profile of $\rho^0$ than $\rho^+$ cells (Figs 2B and 3B). We attributed this to the slower recovery of $\Delta\Psi m$ in $\rho^0$ cells that lack a functional ETC. To explore this aspect further, we monitored cell cycle progression in $\rho^+$ cells lacking Cox4, a nuclear-encoded subunit of complex IV. Unperturbed *cox4$\Delta$* $\rho^+$ cells were able to maintain a $\Delta\Psi m$ that did not significantly differ from that of WT $\rho^+$ cells (Figs 4A and S3B and C). Accordingly, deletion of the *COX4* gene did not have a pronounced effect on the cell cycle profile of unsynchronized $\rho^+$ cells (Fig 4B and C). However, *cox4$\Delta$* $\rho^+$ cells exhibited a somewhat delayed transition from G1 to S in synchrony experiments (Figs 4D and S3D) and a stronger, more sustained cell cycle response to BAM15 than WT or *ATP1-111* $\rho^+$ cells, indicating that a functional ETC aids in timely recovery from uncoupling (Figs 4F and S3F). In line with this notion, all three $\rho^0$ strains, including the *ATP1-111* $\rho^0$ with the restored $\Delta\Psi m$, failed to recover from the BAM15-induced accumulation of G1 cells within the 120-min follow-up period, although the percentage of G1 cells was lower in *ATP1-111* $\rho^0$ than in WT or *cox4$\Delta$* $\rho^0$ cells (Figs 4G and S3F).

Based on the findings in Fig 4, we conclude that the delayed G1-to-S phase progression in $\rho^0$ cells can be attributed to a decreased $\Delta\Psi m$. Moreover, the cell cycle defect and the consequent petite phenotype and slow growth of $\rho^0$ cells can be rescued by increasing $\Delta\Psi m$ through the *ATP1-111* mutation, implicating membrane potential—rather than other metabolic adaptations in $\rho^0$ cells—as the main determinant of

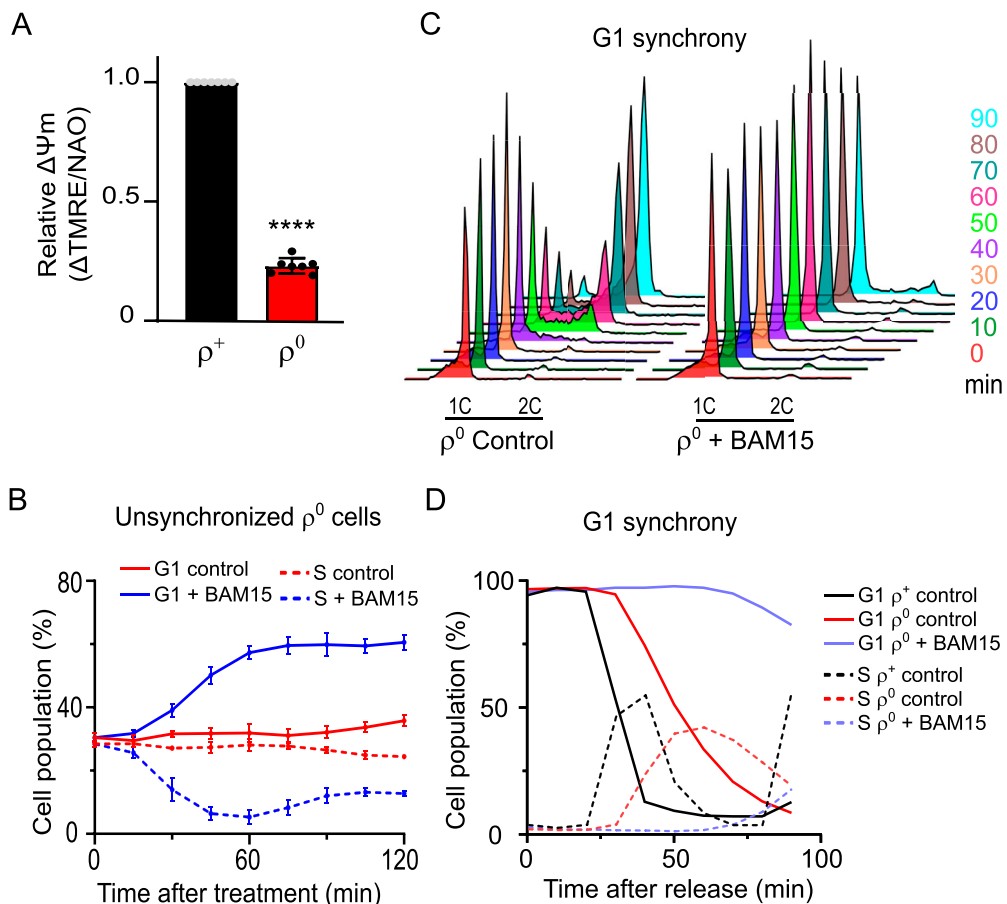

**Figure 3. Uncoupler treatment exacerbates the cell cycle phenotype of $\rho^0$ cells.**
**(A)** ΔΨm normalized to mitochondrial mass was measured in WT (AC403) $\rho^+$ and $\rho^0$ cells as described in the Materials and Methods section. The average of seven independent experiments is shown; error bars represent SD. The groups were compared using one-way ANOVA. ****$P <$ 0.0001. The same data are shown in Fig 4A. **(B)** Percentage of G1 phase (*solid lines*) and S phase (*dashed lines*) cells in early logarithmic phase cultures of WT $\rho^0$ cells left untreated or treated with 30 $\mu$M of BAM15. Aliquots were harvested upon the addition of the drug (0 min) and every 15 min thereafter. Values represent the average of five independent experiments, and error bars indicate SD. Representative DNA histograms are shown in Fig S2C. **(C)** DNA histograms of WT (AC402) $\rho^0$ cells synchronized in G1 with 10 $\mu$g/ml $\alpha$-factor and released into media with or without 30 $\mu$M BAM15. Aliquots were sampled upon release (time 0) and every 10 min thereafter. **(D)** Quantification of G1 phase (*solid lines*) and S phase (*dashed lines*) cells in the experiment presented in Fig 3C (untreated and BAM15-treated $\rho^0$ cells); the $\rho^+$ control data are from Fig 2C. Representative data from a single experiment are shown. See also Fig S2.

the cell cycle delay. Accordingly, recovery from transient uncoupler-induced cell cycle delay is expedited by a functional ETC.

## Altered ROS is not the signal for the G1-to-S phase progression delay in $\rho^0$ cells

Mitochondria are a major source of ROS, and mitochondrial dysfunction is often associated with increased levels of intracellular ROS (3). Furthermore, ROS are documented to influence cell proliferation (1). Therefore, we wanted to determine whether ROS levels play a role in the $\rho^0$ G1-to-S phase progression delay. Rather than increasing the percentage of cells in G1, however, $H_2O_2$ treatment primarily caused the accumulation of WT $\rho^+$ cells in the early S phase, as previously reported by others (Figs 5A and B and S4A) (23). In contrast, decreasing ROS by treatment of WT $\rho^+$ cells with the antioxidants N-acetylcysteine (NAC) or reduced glutathione (GSH) caused a transient accumulation of cells in the G1 phase along with a decrease in the percentage of cells in the S phase (Fig 5C, left panel; Fig S4B). A similar but more sustained effect of treatment with either NAC or GSH was observed in WT $\rho^0$ cells (Fig 5C, right panel; Fig S4B). On their own, these data would suggest that neutralizing ROS by antioxidant treatment inhibits G1-to-S transition. However, measurement of ΔΨm in antioxidant-treated $\rho^+$ cells revealed considerable effects of the antioxidants on ΔΨm, with a

decrease comparable to that after uncoupler treatment (Figs 5D and S4C and D). Similar effects of NAC on ΔΨm have been reported in mammalian cells (24, 25). The impact of antioxidant treatment on *S. cerevisiae* cell cycle progression observed in Fig 5C may therefore be indirect and attributable to decreased ΔΨm rather than lower ROS levels per se.

To further explore the relationship between ROS, ΔΨm, and cell cycle progression, we gauged the levels of oxidative stress in our strains of interest by measuring lipid peroxidation, a reliable indicator of cellular oxidative damage (26, 27). This analysis revealed comparable levels of lipid peroxidation in all six strains, while accurately detecting an increase in lipid peroxidation in control samples treated with $H_2O_2$ (Figs 5E and S4E). Also, levels of mitochondrial superoxide as indicated by MitoSOX Red fluorescence were comparable across our strains of interest (Fig 5F; see Fig S4F for controls). Based on these results, we conclude that the cell cycle phenotype observed in $\rho^0$ cells does not correlate with the cellular levels of oxidative stress and that the rescued cell cycle phenotype in *ATP1-111* $\rho^0$ cells cannot be attributed to normalized ROS levels. Taken together, the findings of this study confirm that diminished ΔΨm, rather than ROS or defective mitochondrial ATP synthesis, underlies the G1-to-S transition delay observed in $\rho^0$ cells and that loss of ΔΨm inhibits cell cycle progression even in $\rho^+$ cells.

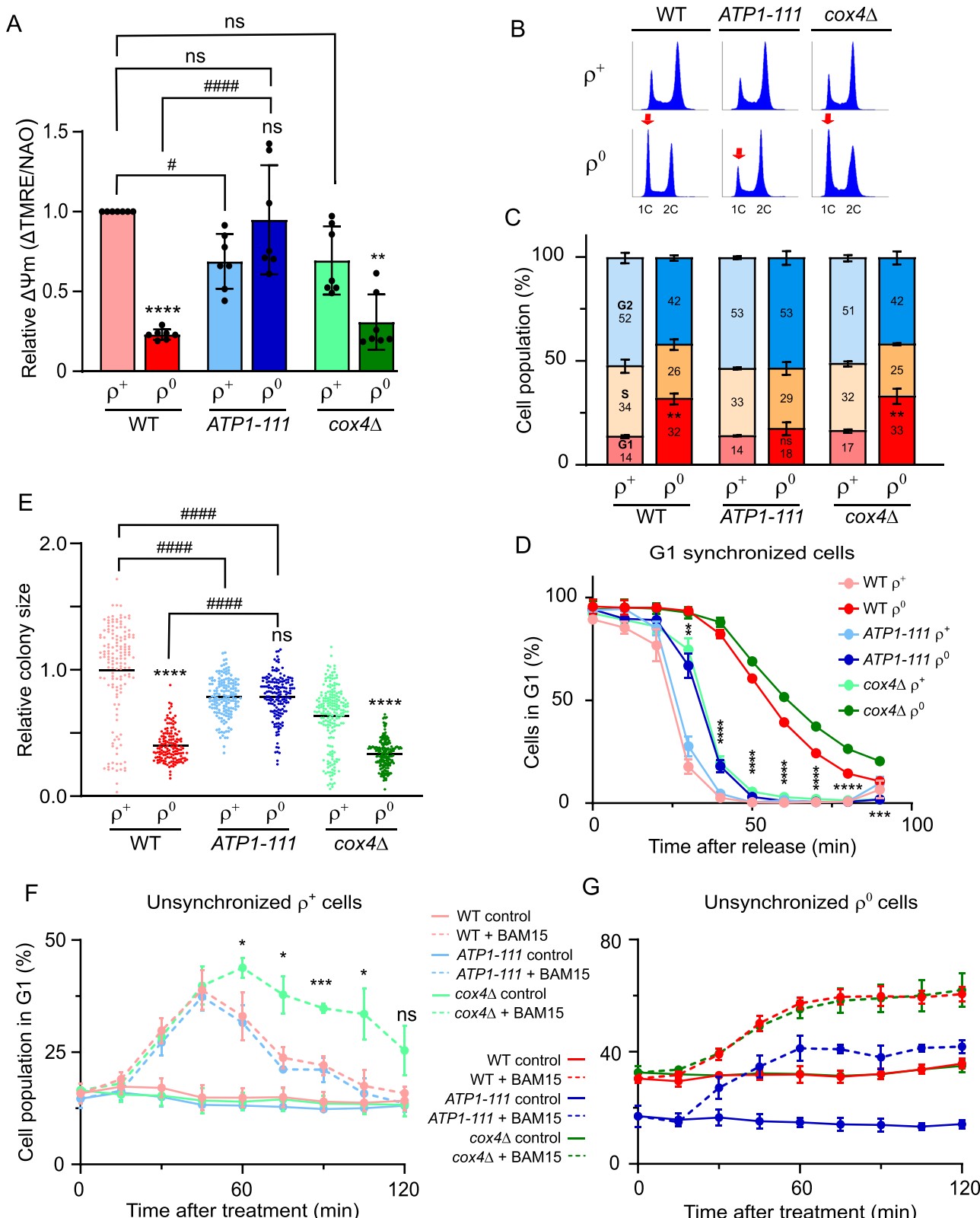

**Figure 4. Cell cycle delay can be rescued by increasing the ΔΨm of ρ⁰ cells.**
**(A)** ΔΨm normalized to mitochondrial mass was measured in WT (AC403), *ATP1-111*, and *cox4Δ* ρ⁺ and ρ⁰ cells as described in the Materials and Methods section. Histograms of TMRE and NAO fluorescence are presented in Fig S3B and C. The average of seven independent experiments is shown; error bars represent SD. The groups were compared by one-way ANOVA. ****$P < 0.0001$, **$P < 0.01$, ns $P > 0.05$ compared with respective ρ⁺; ####$P < 0.0001$, #$P < 0.05$, ns $P > 0.05$ compared with the indicated

## ΔΨm modulates G1/S transition through cell size

The G1/S transition is controlled by a late-G1 checkpoint called *Start*, passage through which requires attainment of a critical cell size and commits cells to cell division (28, 29). In brief, progression through G1 is driven by the G1 cyclin Cln3 in complex with the cyclin-dependent kinase Cdk1 that phosphorylates and partly inactivates the transcriptional inhibitor Whi5, leading to the induction of a transcriptional program encompassing over 200 genes that promotes G1/S transition (30). This transcriptional wave is controlled by two transcription factors, SBF and MBF. SBF regulates the expression of genes involved in cell morphogenesis and the timing of cell cycle commitment such as those encoding the G1 cyclins Cln1 and Cln2. MBF targets, including the S phase cyclins Clb5 and Clb6, drive DNA replication and repair (31). Once expressed, the association of Cln1/2 with Cdk1 promotes features such as bud emergence and pheromone resistance, and Clb5/6-Cdk1 complexes induce DNA replication. A prominent model for how *Start* transition is coupled to attainment of a critical cell size involves the growth-dependent dilution of the Whi5 repressor (32, 33), although other mechanisms have also been proposed (e.g., Refs. (34, 35)). Moreover, cell size, G1/S transcription, and S phase entry are influenced by additional factors and pathways including the cAMP/protein kinase A, TOR, and Snf1/AMPK pathways, the full extent and interaction of which are currently not understood (36). In fact, passage through *Start* is regulated by the integration of multiple internal and external signals including, but not limited to, cell size, nutrient availability, and stress (37).

To gain insight into the mechanisms connecting mitochondrial metabolism and ΔΨm to *Start* passage, we first explored the cell size of WT versus *ATP1-111 ρ⁺* and *ρ⁰* cells. Given that *Start* passage is coupled to cell size, we asked whether the rescued G1/S transition in *ATP1-111 ρ⁰* cells is merely a consequence of larger cell size and thus a shorter "pre-*Start*" G1 period compared with WT *ρ⁰* cells, or whether it is explained by changes to some other aspect regulating *Start* passage, as is observed upon switching to non-fermentable carbon sources when the critical cell size required for G1/S transition decreases (38). Microscopy-based measurement of the area of G1 phase (=unbudded) cells revealed that although WT and *cox4Δ ρ⁰* cells were smaller than their *ρ⁺* counterparts, the size of *ATP1-111 ρ⁰* cells was comparable to that of *ATP1-111 ρ⁺* and WT *ρ⁺* cells (Figs 6B and S5A). Thus, the rescued G1/S transition in *ATP1-111 ρ⁰* cells with a partially recovered ΔΨm may be explained by their larger size and the subsequently shorter pre-*Start* G1 period.

The larger size of *ATP1-111 ρ⁰* cells is expected to lead to a faster activation of G1 transcription because of earlier dilution of the transcriptional inhibitor Whi5 (Fig 6A). We analysed the timing of the G1/S transcriptional wave in WT versus *ATP1-111 ρ⁺* and *ρ⁰* cells released from α-factor synchrony, an experimental approach previously used by others (39, 40). As expected (41), no fluctuation of the levels of the non-cycling G1 cyclin *CLN3* was observed after release, and transcript levels of *CLN3* or *WHI5* did not significantly differ across the analysed strains (Figs 6C and S5B and C). In contrast, the induction of the late-G1 cyclins *CLN1* and *CLN2* was delayed in WT *ρ⁰* cells as compared to WT *ρ⁺* cells. A similar but somewhat weaker tendency was observed for the S phase cyclins *CLB5* and *CLB6* (Fig S5C). Unexpectedly, and in discordance with the rescued G1/S transition phenotype, the transcriptional pattern of these cyclins was not restored in *ATP1-111 ρ⁰* cells (Fig 6C; compare *ATP1-111 ρ⁰* with WT *ρ⁰*). The rescued cell cycle phenotype of *ATP1-111 ρ⁰* cells does therefore not appear to be explained by an earlier accumulation of *CLN1/2* or *CLB5/6* gene products.

Taken together, the findings in Fig 6 indicate that *ATP1-111 ρ⁰* cells are larger in size than WT *ρ⁰* cells, likely explaining their rescued cell cycle phenotype. However, the rescued timing of their *Start* passage appears to involve improvements downstream of G1 and S cyclin transcription. Signalling through numerous pathways, including the PKA, TOR, or Snf1/AMPK pathways, is known to impinge on *Start* timing, and their involvement, and that of any Cdk inhibitors acting on the Cln1/2-Cdk1 complex, should be addressed in future work to gain further insight into cell cycle regulation in cells with mitochondrial dysfunction.

# Discussion

The complexes of the ETC and ATP synthase are encoded on both cellular genomes, making retrograde communication from the mitochondria to the nucleus a prerequisite for mitochondrial biogenesis and the correct assembly of these critical complexes. Mito-cellular signalling is also required to trigger any compensatory responses that allow the cell to adjust to mitochondrial dysfunction. Therefore, inter-organellar communication initiated in the mitochondria is crucial under both physiological and pathological conditions.

A number of studies have implicated signals of mitochondrial status in the control of cell cycle progression. Mitochondrial dysfunction brought about by ETC mutations was shown to delay G1-to-S transition in *Drosophila melanogaster* imaginal disc cells (42, 43). Interestingly, these studies found different mitochondrial cues to be responsible for triggering the G1/S cell cycle checkpoint

---

strain. WT *ρ⁰* did not significantly differ from *cox4Δ ρ⁰*; WT *ρ⁺* versus *cox4Δ ρ⁰*: ####; WT *ρ⁰* versus *ATP1-111 ρ⁺*: ##. **(B)** Representative DNA histograms of WT (AC403), *ATP1-111*, and *cox4Δ ρ⁺* and *ρ⁰* cells grown to an early logarithmic phase in YPDA. **(B, C)** Quantification of the percentage of cells in the G1, S, or G2 phase in panel (B). Values represent the average of at least three independent experiments, and error bars indicate SD. The two-tailed *t* test was performed to determine statistical significance between the G1 populations of each strain's *ρ⁺* and *ρ⁰* variants. **$P < 0.01$, ns $P > 0.05$. **(D)** Quantification of the percentage of G1 cells in WT (AC402), *ATP1-111*, and *cox4Δ ρ⁺* and *ρ⁰* cells after release from G1 synchrony. Representative histograms are shown in Fig S3D. Values represent the average of at three independent experiments, and error bars indicate SD. The percentage of cells in the G1 phase between *ATP1-111 ρ⁰* cells and WT *ρ⁰* cells was compared by a two-tailed *t* test. ****$P < 0.0001$, ***$P < 0.001$, **$P < 0.01$. **(E)** Quantification of the colony sizes of WT (AC403), *ATP1-111*, and *cox4Δ ρ⁺* and *ρ⁰* cells grown on the same YPDA plate for 48 h; the average area of WT *ρ⁺* colonies was set to 1. The groups were compared by one-way ANOVA. ****$P < 0.0001$, ns $P > 0.05$ compared with the respective *ρ⁺*; ####$P < 0.0001$ compared with the indicated strain. **(F, G)** Percentage of WT (AC403), *ATP1-111*, and *cox4Δ ρ⁺* (F) and *ρ⁰* cells (G) in the G1 phase in untreated cultures (*solid lines*) or after treatment with 20 µM of BAM15 (*dashed lines*). The average of three independent experiments is shown; error bars represent SD. The percentage of cells in the G1 phase between WT and *cox4Δ ρ⁺* cells at specific timepoints was compared by a two-tailed *t* test. ***$P < 0.001$, *$P < 0.05$, ns $P > 0.05$. See also Fig S3.

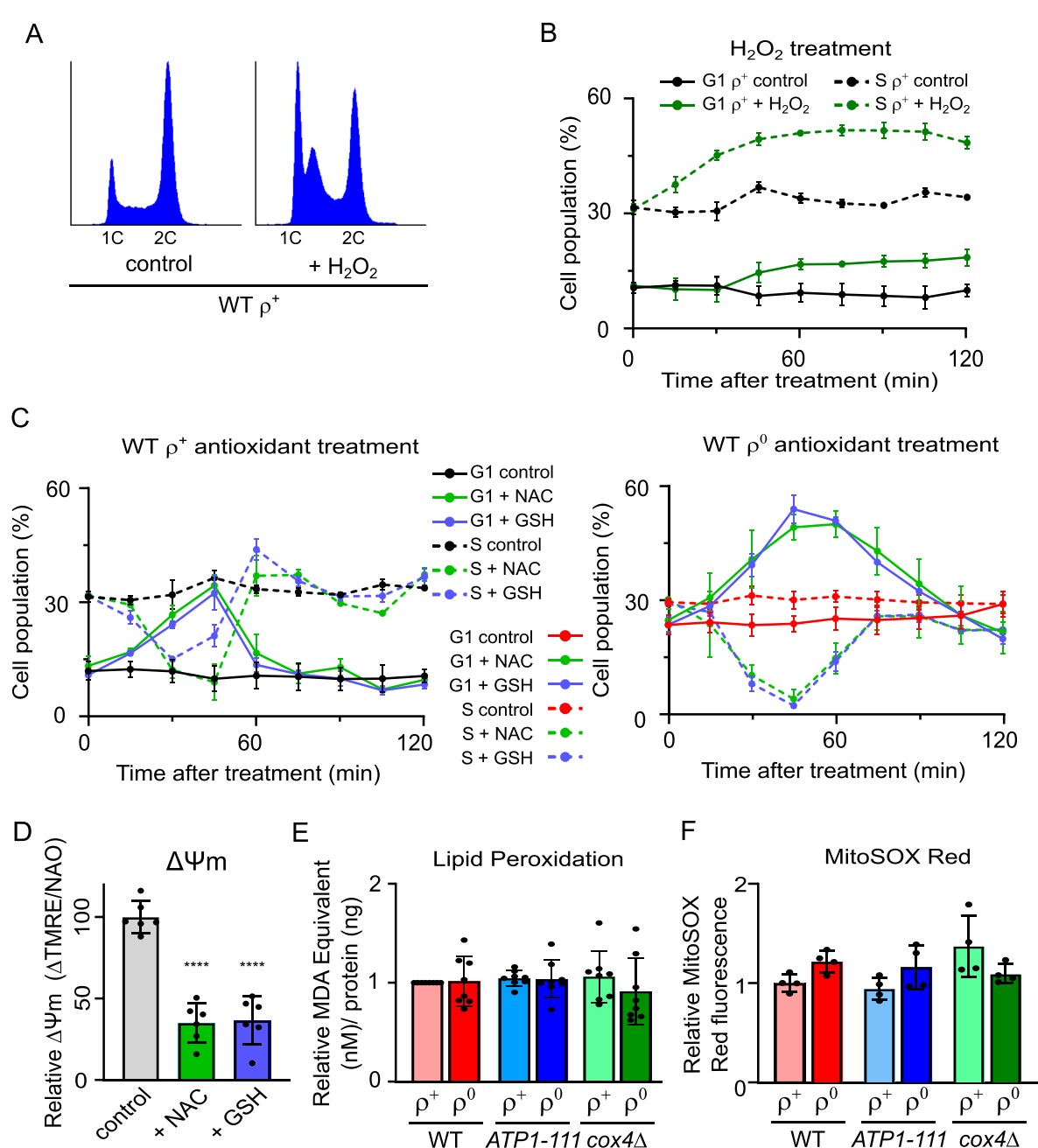

**Figure 5. Levels of oxidative stress do not correlate with the extent of the G1-to-S delay.**
**(A)** Representative DNA histogram of WT (AC402) $\rho^+$ cells grown until an early logarithmic stage and treated with 400 $\mu M$ $H_2O_2$ for 60 min. Additional timepoints are shown in Fig S4A. **(B)** Quantification of G1 phase (*solid lines*) and S phase (*dashed lines*) cells in the experiment presented in Fig 5A and three additional experiments; error bars indicate SD. **(C)** Quantification of % of G1 phase (*solid lines*) and S phase (*dashed lines*) cells in WT $\rho^+$ (left panel) and WT $\rho^0$ (right panel) strains treated with 30 mM NAC or 20 mM GSH for 2 h. Values represent the average of three experiments, and error bars indicate SD. Representative DNA histograms are shown in Fig S4B. **(D)** $\Delta\Psi m$ normalized to mitochondrial mass was measured in WT (AC402) cells left untreated and treated with either 30 mM NAC or 20 mM GSH for 10 min. The average of six independent experiments is shown; error bars represent SD. The control and the treated samples were compared by one-way ANOVA. ****$P < 0.0001$. Raw TMRE intensity data and related controls are shown in Fig S4C and D. **(E)** Quantification of lipid peroxidation in WT (AC402), *ATP1-111*, and *cox4Δ* $\rho^+$ and $\rho^0$ cells grown on YPDA until an early logarithmic phase. Values are expressed as MDA equivalent (nM) per ng total protein. The average of eight independent experiments is shown; error bars represent SD. There is no significant difference between sample groups after comparison by one-way ANOVA. **(F)** Quantification of MitoSOX Red fluorescence as an indicator of mitochondrial superoxide in WT (AC402), *ATP1-111*, and *cox4Δ* $\rho^+$ and $\rho^0$ cells grown until an early logarithmic stage. The average of four independent experiments is shown; error bars represent SD. There is no significant difference between sample groups after comparison by one-way ANOVA. See also Fig S4.

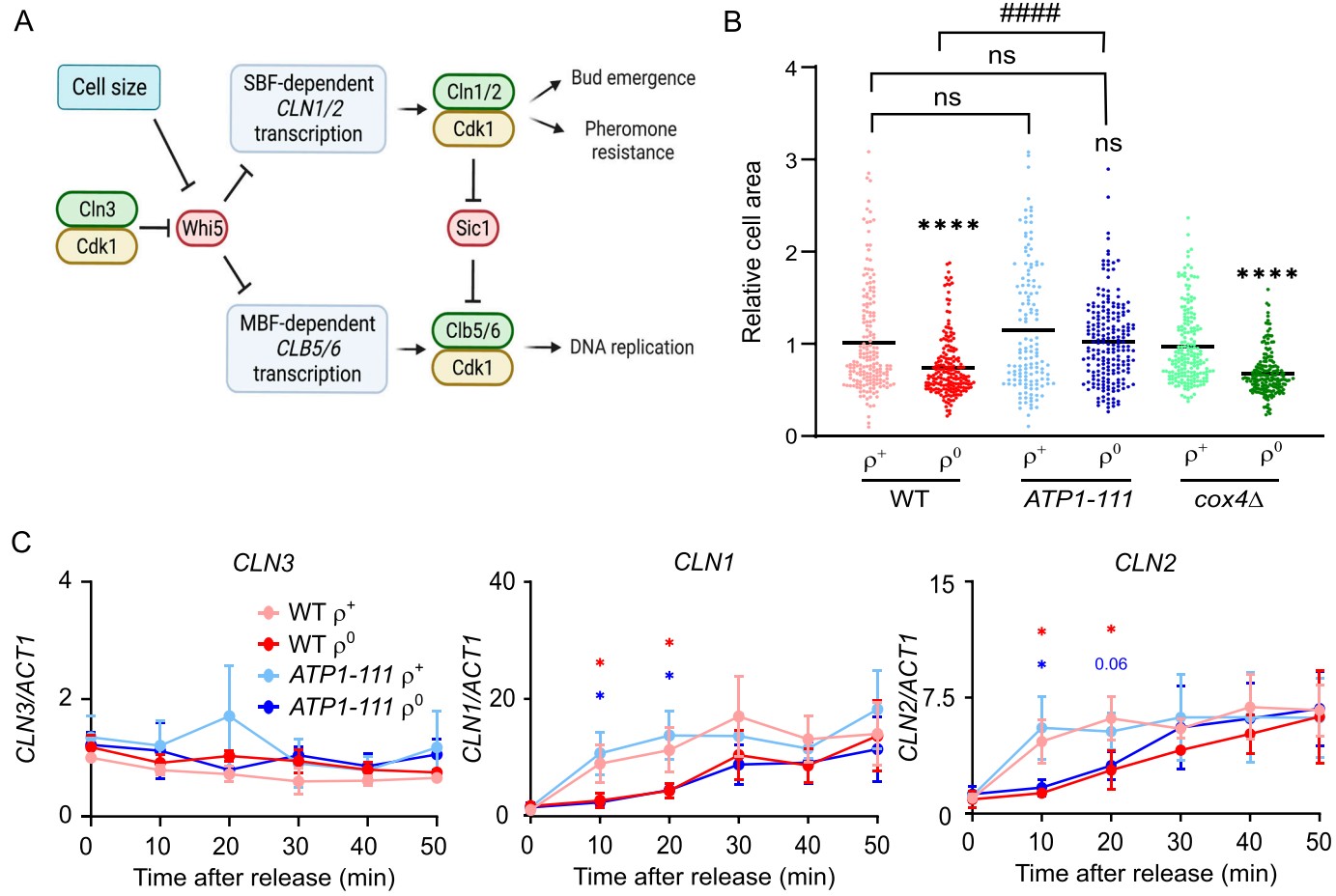

**Figure 6. ΔΨm modulates G1/S transition through cell size.**
**(A)** Brief schematic of *Start* passage control. The Cln3-Cdk1 complex partly inactivates the transcriptional repressor Whi5, relieving the repression of SBF- and MBF-dependent genes such as *CLN1/2* and *CLB5/6*, respectively. Cln1/2-Cdk1 activity, which is further modulated by Cdk inhibitors such as Cip1 and Far1 (*not shown*), drives G1/S-associated changes such as budding and pheromone resistance. Cln1/2-Cdk1 also inactivates the Sic1 inhibitor of Clb5/6-Cdk1, allowing the latter to activate DNA replication. Numerous other pathways impinge on this control; reviewed in Refs. (31, 36, 37). **(B)** Area of cross-sections of G1 (=unbudded) cells stained with trypan blue; the size of WT $\rho^+$ cells was set to 1. An average of 180 G1 cells were counted per group; the line indicates the mean value. The groups were compared by one-way ANOVA. ****$P < 0.0001$, ns $P > 0.05$ compared with the respective $\rho^+$; ####$P < 0.0001$ compared with the indicated strain. **(C)** Relative gene expression analysis of the *CLN3*, *CLN1*, and *CLN2* transcripts at the indicated timepoints after release of WT or *ATP1-111* $\rho^+$ and $\rho^0$ cells from α-factor synchrony. Values were normalized to *ACT1* at each timepoint, and the *CLN/ACT1* ratio of WT $\rho^+$ at 0 min was set to 1. The average of three independent experiments is shown; error bars indicate the standard error of the mean. Asterisks indicate $P < 0.05$ in a two-tailed $t$ test comparing WT $\rho^+$ versus $\rho^0$ (*red asterisks*) or *ATP1-111* $\rho^+$ versus $\rho^0$ (*blue asterisks*). See Fig S5B for a representative cell cycle profile of released cells.

depending on the underlying ETC subunit mutation: although a complex IV mutation that decreased ATP levels signalled through AMPK and p53 to prevent S phase entry, a complex I mutation induced high ROS and signalled through the Foxo/p27 pathway. These findings highlight the fact that the signals and pathways that modulate the cell cycle in response to mitochondrial dysfunction can vary even within a single type of cell.

In *S. cerevisiae*, the identity of the mitochondrial signal that triggers the accumulation of respiratory-deficient $\rho^0$ cells in G1 has remained unknown. In the current study, we rule out the two cues of mitochondrial dysfunction identified in *Drosophila*—loss of mitochondrial ATP production and high ROS—as triggers of the yeast cell cycle delay. Instead, we show that the G1-to-S transition delay observed in $\rho^0$ cells is caused by decreased ΔΨm. Accordingly, the cell cycle phenotype of $\rho^0$ cells could be recovered by increasing ΔΨm,

indicating that low ΔΨm rather than respiratory deficiency per se restricts cell cycle progression in $\rho^0$ cells (Figs 3–5). Dissipation of ΔΨm in $\rho^+$ cells induced a similar but transient G1-to-S delay (Fig 2). These results therefore establish mitochondrial membrane potential as a general modulator of cell cycle progression in *S. cerevisiae* cells.

In mammalian cells, the ΔΨm has been reported to fluctuate over the course of the cell cycle, with the highest ΔΨm, mitochondrial $O_2$ consumption, and ATP synthesis measured just before G1-to-S transition (44, 45). Furthermore, (45) elegantly demonstrated that the boost in ΔΨm in late-G1 cells was required for S phase entry (45). The ΔΨm therefore appears to regulate cell cycle progression even in higher eukaryotes. In addition, the mitochondrial membrane potential is known to govern, for example, mitochondrial dynamics and quality control through mitophagy (46, 47), highlighting its role as a central readout of mitochondrial status.

**Table 1.** Yeast strains used in this study.

| Strain | Genotype | Source |
|---|---|---|
| AC402 | MATa ade2-1 CAN1+ his3-11,15 leu2-3,112 trp1-1 ura3-1 RAD5+ [ρ⁺] | (51) |
| AC403 | MATα | (52) |
| AC402 ρ⁰ | MATa [ρ⁰] | This study |
| AC403 ρ⁰ | MATα [ρ⁰] | This study |
| PW66-1C | MATα cox4Δ::NatMX [ρ⁺] | This study |
| PW66-1C ρ⁰ | MATα cox4Δ::NatMX [ρ⁰] | This study |
| PW66-1A | MATa cox4Δ::NatMX [ρ⁺] | This study |
| PW66-1A ρ⁰ | MATa cox4Δ::NatMX [ρ⁰] | This study |
| PW72 | MATα ATP1-111 [ρ⁺] | This study |
| PW72 ρ⁰ | MATα ATP1-111 [ρ⁰] | This study |
| PW130-1A | MATa ATP1-111 [ρ⁺] | This study |
| PW130-1A ρ⁰ | MATa ATP1-111 [ρ⁰] | This study |
| PW104-A | MATa rim1Δ::KanMX4 | This study |
| PW111 | MATa mip1Δ::KanMX4 | This study |
| PW114 | MATa mgm101Δ::KanMX4 | This study |

All strains are isogenic to W303 and derivatives of W4069-4c (AC402); only genotypes differing from AC402 are listed.

**Table 2.** Oligonucleotides used in the study.

| Name | Sequence |
|---|---|
| Actin forward | 5′-CAT GAT ACC TTG GTG TCT TGG-3′ |
| Actin reverse | 5′-GTA TGT GTA AAG CCG GTT TTG-3′ |
| Cox1 forward | 5′-CTA CAG ATA CAG CAT TTC CAA GA-3′ |
| Cox1 reverse | 5′-GTG CCT GAA TAG ATG ATA ATG GT-3′ |
| CLN1 forward | 5′-ACA TTG ACC ATT CAT CGC CG-3′ |
| CLN1 reverse | 5′-AGC GGA TGA TGA GTT GGG AA-3′ |
| CLN2 forward | 5′-TGT CTC TGG TTG GCT GCT AA-3′ |
| CLN2 reverse | 5′-AGA CCT GAC CAT CAC CAC AG-3′ |
| CLN3 forward | 5′-CCA CCC TTT GCT TTC ACT CC-3′ |
| CLN3 reverse | 5′-AGG AGT TAG TGG ACT TGG CC-3′ |
| CLB5 forward | 5′-GGA GAG AAC CAC GAC CAT GA-3′ |
| CLB5 reverse | 5′-AGA ATC CTG AAC CTG CTG CT-3′ |
| CLB6 forward | 5′-GGC GCT GTT AAT TGA CTG GT-3′ |
| CLB6 reverse | 5′-AGG CAA TGA ACA GGC AAG TG-3′ |
| WHI5 forward | 5′-CCA CGT CGC TAT CAC AAC AG-3′ |
| WHI5 reverse | 5′-TCG GTG TTG TTG CTT CTT GG-3′ |
| ACT1 forward (RT–PCR) | 5′-CGA ATT GAG AGT TGC CCC AG-3′ |
| ACT1 reverse (RT–PCR) | 5′-CAA GGA CAA AAC GGC TTG GA-3′ |

We further found that the improved ΔΨm of *ATP1-111* ρ⁰ cells resulted in a larger cell size, which likely explains the faster G1/S transition as cells need less time to reach the critical cell size required for *Start*. Based on the Whi5 dilution model, a larger cell size is predicted to lead to an earlier activation of G1/S transcription, but this was not observed in *ATP1-111* ρ⁰ cells (Fig 6).

Another factor connecting cell size to the timing of *Start* is Cln3 concentration (e.g., Ref. (34)). Although not studied here, the position of Cln3 upstream of *CLN1/2* and *CLB5/6* expression, which were not rescued, makes changes to Cln3 activity an unlikely explanation for the faster G1/S transition in *ATP1-111* ρ⁰. However, multiple additional pathways, for example, some involved in nutrient sensing and stress, are known to modulate *Start* (36, 37, 38). Future investigations should seek to identify the mechanisms that connect ΔΨm changes to the cell cycle machinery. This and many related aspects of the mito-cellular communication can in the long term help advance our understanding of mitochondrial disease.

Mitochondrial dysfunction underlies a diverse group of diseases that is caused by defects in mitochondrially localized proteins encoded on either the nuclear or the mitochondrial genome. The common feature of mitochondrial diseases is a defective function of the respiratory chain and/or ATP synthase that leads to insufficient mitochondrial ATP synthesis. In most cases, mitochondrial disorders also involve decreased ΔΨm (48). However, not all symptoms of mitochondrial disease can be directly attributed to changes in energy metabolism, as some stem from (mal)adaptive responses to the mitochondrial dysfunction. For example, dysregulated immune signalling triggered by mtDNA instability was shown to aggravate the metabolic dysfunction in a mouse model with an increased mtDNA mutation load (49). Similarly, patient fibroblasts carrying the common m.3242 A>G mtDNA point mutation exhibited constitutive activation of the PI3K-Akt-mTORC1 signalling axis, the inhibition of which partly improved mitochondrial function (50). These examples illustrate the importance of understanding the various forms of mito-cellular crosstalk and the possible therapeutic potential of inhibiting signalling that in some cases may be more detrimental than beneficial. It remains to be seen whether inhibition of normal cell cycle progression in response to

decreased ΔΨm contributes to any of the symptoms of mitochondrial disease. If so, targeting the factors that mediate this signalling may provide a potential avenue for treating that subset of disease symptoms.

# Materials and Methods

### Yeast strains and growth conditions

Unless otherwise indicated, all *S. cerevisiae* strains used in this study are congenic to W4069-4c, which is in a W303 background (51), and are listed in Table 1. To ensure fitness before each experiment, all strains were streaked out from glycerol stocks onto a YPDA solid medium (1% yeast extract, 2% peptone, 50 mg/l adenine, 2% glucose, and 2% agar). Unless otherwise indicated, yeast cells were grown in YPDA at 30°C with shaking at 180 rpm.

$P^0$ strains lacking mtDNA were generated by growth in the YPDA medium supplemented with 25 μM ethidium bromide for 4–15 d depending on the strain (12). Loss of mtDNA was confirmed by lack of growth on the YPDA medium (1% yeast extract, 2% peptone, 50 mg/l adenine, and 3% glycerol) in combination with the absence of amplification of a region of the mitochondrially encoded *COX1* gene by quantitative real-time PCR in samples where a region of the nuclear-encoded *ACT1* gene did amplify. Real-time PCRs contained 0.2 μM forward and reverse primers targeting *COX1* or *ACT1* (Table 2) and 10 μl 2x SyGreen Mix (PCR Biosystems), and were run on a LightCycler 96 instrument (Roche) using the following program: 95°C for 180 s, 45 cycles of (95°C for 10 s, 56°C for 10 s, and 72°C for 1 s with signal acquisition), and melting curve (95°C for 5 s, 65°C for 60 s, and heating to 97°C at 1 s with continuous signal acquisition).

The strain bearing the *ATP1-111* mutation was generated by introduction of the *ATP1* variant carrying a T>G substitution at position 331 from a pRS316 integration vector as previously described (19, 22). Briefly, the *ATP1-111* insert in pRS306 was excised with BamHI and XbaI and re-cloned into the pRS316 integration vector. Yeast cells were transformed using a standard lithium acetate procedure (53), and the presence of the mutation in cells that grew in a medium supplemented with 5-FOA was confirmed by sequencing.

Deletion strains were constructed by replacing the entire ORF with the *kanMX*4 or *natMX* cassette (54). Gene deletion was confirmed by growth in geneticin or nourseothricin, respectively, and by PCR using both a primer pair that flanks the ORF and a pair internal to it.

### Colony size and growth curves

Yeast strains grown in a YPDA liquid medium for at least 24 h were counted using Neubauer Improved Hemocytometer and adjusted to 100 cells per 100 μl dH₂O. 100 μl of the cell suspension was spread on a single-partitioned YPDA plate and incubated at 30°C for 48 h before imaging on a ChemiDoc imaging system (Bio-Rad). The area of all colonies, except for ones with another colony attached to them, was analysed using Fiji (55), and individual values were normalized to the average of all WT $ρ^+$ cell colonies.

For analysis of growth in liquid media, exponentially growing cultures were re-inoculated into fresh YPDA liquid media with a starting OD$_{600}$ = 0.1. OD$_{600}$ readings were recorded every hour by spectrophotometry.

### Cell cycle analysis by flow cytometry

Cells grown overnight in YPDA were diluted to a fresh medium to a starting OD$_{600}$ of 0.1. Cells were harvested at OD$_{600}$ 0.35–0.5, fixed with cold 70% ethanol, and incubated at 4°C overnight. The fixed cells were washed with distilled water once, resuspended in 50 mM Tris–HCl, pH 7.5, with 15 mM NaCl, and treated with 2 mg/ml RNase A at 37°C overnight. The next day, samples were treated with 1.8 mg/ml proteinase K at 50°C for 1 h, spun down, and resuspended in 50 mM Tris, pH 7.5. An aliquot was added to 50 mM Tris, pH 7.5, containing a 1:10,000 dilution of SYBR Green nucleic acid stain (Invitrogen). The cell suspension was sonicated for 10 s with an amplitude of 20% before analysis (Q500; QSonica). DNA content was detected in the FL1 channel using the Cytomics FC500 (Beckman Coulter). Data analysis and quantification of cells in G1, S, and G2 phases was performed using FCS Express 7 Flow (De Novo Software). The y-axis scale of the cell cycle histograms was adjusted to fit the highest cell count.

For time-course analysis of unsynchronized cells, 30 ml culture (with a starting OD$_{600}$ of 0.1) was grown to an OD$_{600}$ of 0.4–0.6 and split into two portions: the specified drug added to one portion, whereas the other served as a control. Cell aliquots were harvested and immediately fixed every 15 min for 2 h. Time 0 indicates the addition of the drug.

For analysis of synchronized cells, cultures were synchronized in G1 by the addition of 5 μg/ml α-factor pheromone every hour for 2 h. For G2 synchronization, cells were treated with 10–25 μg/ml nocodazole for 2 h. Cells were washed twice with one volume of ice-cold distilled water and released from synchrony by resuspension in the fresh medium. Cells were harvested and immediately fixed every 10 min for 90–120 min after release.

### Mitochondrial membrane potential

Cells grown overnight in YPDA were diluted in the fresh medium to a starting OD$_{600}$ of 0.1 and grown for 24 h; 1 ml of the culture was harvested and washed twice with one volume of PBS buffer (0.14 M NaCl, 0.0027 M KCl, and 0.010 M phosphate buffer, pH 7.4). Cells were reconstituted in 5 ml PBS to an OD$_{600}$ of 0.05. A 1 ml aliquot was treated with 2 μM tetramethylrhodamine methyl ester perchlorate (TMRE) (Molecular Probes) and incubated at 37°C for 30 min. As a control, another aliquot of the cell suspension was treated with the same concentration of TMRE for 20 min followed by the addition of 20 μM carbonyl cyanide 3-chlorophenyl hydrazone (CCCP) for 10 min to collapse the membrane potential. When different strains were compared, mitochondrial mass of each strain was estimated by adding 250 nM NAO (Invitrogen) to a third aliquot of the cell suspension and incubated at 37°C for 30 min. Both mitochondrial membrane potential and mitochondrial mass were determined by flow cytometry on a Beckman Coulter Cytomics FC500. ΔΨm was calculated by subtracting the TMRE fluorescence of the untreated

and uncoupled sample, and, when comparing across strains, normalized for mitochondrial mass (19).

## Lipid peroxidation analysis and detection of mitochondrial ROS

Lipid peroxidation was determined by measurement of the amount of thiobarbituric acid reactive substance using QuantiChrom Thiobarbituric Acid Reactive Substance Assay Kit (BioAssay Systems) following the manufacturer's instructions. The concentration of malondialdehyde (MDA) was measured on a Synergy H4 hybrid microplate reader (BioTek) with a fluorescence intensity of ($\lambda_{em/ex}$ = 530 nm/550 nm). Lipid peroxidation was expressed as moles of MDA equivalent equilibrated to the total amount of protein (nM of MDA equivalent/ng protein). Protein concentration was measured using the Quick Start Bradford protein assay (Bio-Rad).

Detection of mtROS was as previously described with some modifications (56). Briefly, overnight cell cultures were diluted to an $OD_{600}$ of 0.1 in a fresh YPDA medium and grown until an $OD_{600}$ of 0.5–0.6. Cells were harvested and washed twice with PBS. Cells were resuspended in PBS to an $OD_{600}$ of 0.5 and incubated with 5 $\mu$M MitoSOX Red (Invitrogen) at 30°C for 15 min before analysis by flow cytometry.

## Gene expression analysis (RT–qPCR)

Cells released from $\alpha$-factor–induced G1 synchrony were harvested every 10 min and frozen in liquid nitrogen. Cells were lysed using 0.5 mm Glass Beads (BioSpec) in a FastPrep-24 bead beater (MP Biomedical), and RNA was extracted using RNeasy Mini Kit (QIAGEN) according to the manufacturer's instructions. 1 $\mu$g of the harvested RNA was treated with 1U DNase I (Thermo Fisher Scientific), and cDNA was synthesized using SuperScript III Reverse Transcriptase (Invitrogen). Gene expression was analysed in technical duplicates by quantitative real-time PCR using 12.5 ng of cDNA in a 20 $\mu$l reaction containing 0.4 $\mu$M forward and reverse primers (Table 2) and 10 $\mu$l 2x SyGreen Mix (#PB20.14-05; qPCRBIO) in a LightCycler instrument (Roche). The 180 s of preincubation at 95°C was followed by 40 cycles of amplification consisting of 95°C for 10 s, 56°C for 10 s, and 72°C for 1 s with single signal acquisition, and melting curve of 95°C for 5 s, 65°C for 60 s, and 97°C at 0.2°C/sec with continuous signal acquisition. Cq values determined by LightCycler 96 software (Roche) were used to calculate the expression of the cyclin genes relative to actin using the Pfaffl method (57). Statistical comparisons between selected timepoints of individual strains were performed by a standard two-tailed $t$ test in GraphPad Prism (GraphPad Software, Inc.).

## Cell imaging and size analysis

For mitochondrial staining, yeast cells grown in YPDA until an early stationary phase were harvested and washed twice with PBS. Cells were then resuspended in 1 ml PBS at a total cell density of $OD_{600}$ = 1.0. The cell suspension was co-stained with 250 nM NAO (manufacturer) and 250 nM MitoTracker Deep Red (Thermo Fisher Scientific) and incubated at 37°C for 10 min. 10 $\mu$l of the stained cells was mixed with a 10 $\mu$l UltraPure LMP Agarose (37°C, 1% [wt/vol] in

PBS) (Invitrogen) and applied to a standard microscopy glass slide and coverslip (VWR). For yeast cell wall staining, early stationary phase cells were harvested and washed twice with PBS. Cells were then resuspended in 1 ml PBS with a total cell concentration of $OD_{600}$ = 1.0, stained with 20 $\mu$g/ml trypan blue (Invitrogen), and applied to a standard microscopy glass slide and coverslip. A Leica Widefield Thunder microscope equipped with an HC PL FLUOTAR 100x/1.32 OIL PH3 objective, with filters and laser settings for DAPI, FITC, TRITC, and Alexa Fluor 674, was used for all microscopic studies. Images were analysed using Fiji software (55). The area of unbudded (i.e., G1) cells was determined by drawing a region of interest around the stained cell wall and measuring the mean grey value using Fiji.

# Supplementary Information

# Acknowledgements

We thank Dr. Andrei Chabes (Umeå University, Umeå, Sweden) for yeast strains, and Drs. Michal Jazwinski (Tulane University, LA, USA) and Anders Byström (Umeå University, Umeå, Sweden) for plasmids. We acknowledge the Biochemical Imaging Center at Umeå University and the National Microscopy Infrastructure (VR-RFI 2019-00217) for assistance in microscopy, the Chemical Biology Consortium Sweden (CBCS) at Umeå University for access to the Synergy plate reader, Dr. Sushma Sharma for expert technical advice and helpful discussions, and the P. Wanrooij laboratory for critical reading of the article. Fig 6A was made using BioRender. This project was supported by grants from the Swedish Cancer Society (19 0022JIA and 19 0098Pj), the Swedish Research Council (2019-01874), the Swedish Society for Medical Research (S17-0023), the Kempe Foundations (JCK-1830), and the Åke Wiberg Foundation (M20-0132) to PH Wanrooij.

## Author Contributions

CM Gorospe: data curation, formal analysis, validation, investigation, visualization, methodology, project administration, and writing—original draft, review, and editing.
G Carvalho: formal analysis, investigation, methodology, and writing—original draft, review, and editing.
A Herrera Curbelo: formal analysis, investigation, and writing—review and editing.
L Marchhart: formal analysis, investigation, and writing—review and editing.
IC Mendes: investigation and writing—review and editing.
K Niedźwiecka: investigation and writing—review and editing.
PH Wanrooij: conceptualization, data curation, formal analysis, supervision, funding acquisition, validation, investigation, methodology, project administration, and writing—original draft, review, and editing.

## Conflict of Interest Statement

The authors declare that they have no conflict of interest.

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
