## [Reviewer comments · Life Science Alliance]

Life Science Alliance

Mitochondrial membrane potential acts as a retrograde signal to regulate cell cycle progression

Choco Michael Gorospe, Gustavo Carvalho, Alicia Herrera Curbelo, Lisa Marchhart, Isabela Mendes, Katarzyna Niedzwiecka, and Paulina Wanrooij

DOI: <https://doi.org/10.26508/lsa.202302091>

Corresponding author(s): Paulina Wanrooij, Umeå University

Review Timeline:

Submission Date:	2023-04-12
Editorial Decision:	2023-05-22
Revision Received:	2023-08-09
Editorial Decision:	2023-08-28
Revision Received:	2023-08-29
Accepted:	2023-08-30

Transaction Report:

May 22, 2023

Re: Life Science Alliance manuscript #LSA-2023-02091

Dr. Paulina H Wanrooij
Umeå University
Medical Biochemistry and Biophysics
Linnaeus väg 8
Umeå, - 90187
Sweden

Dear Dr. Wanrooij,

Thank you for submitting your manuscript entitled "Mitochondrial membrane potential acts as a retrograde signal to regulate cell cycle progression" to Life Science Alliance. The manuscript was assessed by expert reviewers, whose comments are appended to this letter. We invite you to submit a revised manuscript addressing the Reviewer comments.

Thank you for this interesting contribution to Life Science Alliance. We are looking forward to receiving your revised manuscript.

Sincerely,

B. MANUSCRIPT ORGANIZATION AND FORMATTING:

Reviewer #1 (Comments to the Authors (Required)):

This manuscript is well-written about the relationship between mitochondrial membrane potential and cell cycle delay. Their ways to display experimental evidence are sound and solid for this. In brief, the authors reveal that mitochondrial membrane potential act as retrograde signals which affect the cell cycle progression of cells in the manuscript. They compared mtDNA-deficient rho0 cells with normal cells (rho+), and revealed that the cell cycle of yeast is delayed in G1 when mtDNA is deficient. They also observed how inhibiting or overexpressing various mitochondrial functions including ATP synthesis, oxidative stress regulation, and mitochondrial membrane potential), in which mitochondria are mainly involved to affect the cell cycle. Although, they didn't identify specific pathway to regulate cell cycle delay from mitochondrial membrane potential, this manuscript clearly notes the effect of mitochondrial membrane potential on the cell cycle. I recommend to accept the manuscript as it is.

Reviewer #2 (Comments to the Authors (Required)):

The manuscript entitled "Mitochondrial membrane potential acts as a retrograde signal to regulate cell cycle progression" investigated how the mitochondria-related parameters affect cell cycle progression in wild-type Rho+ and Rho-0 mutant, lacking mitochondrial DNA in yeast. I have carefully examined the manuscript and I would like to commend the authors on their well-written and well-documented manuscript, which effectively utilizes genetic and pharmacological approaches to examine the role of mitochondrial membrane potential in the progression of the cell cycle. Overall, the experimental plans are sound, and the data presented is robust. However, it seems that some of the conclusions reached lack direct experimental evidence, despite their significance:

Using uncouplers, authors demonstrated that a $\Delta\Psi_m$ collapse causes G1-to-S phase transition delay, similar to the one observed in Rho0 cells and concluded the role of mitochondrial membrane potential in cell cycle progression. To ensure a thorough understanding of the underlying mechanisms, it is crucial to eliminate other mitochondria-related parameters for e.g., ROS and ATP in cell cycle progression. For instance, mild uncoupling has been shown to typically lower ROS levels due to increased respiratory activity (DOI: 10.1074/jbc.M408918200), consequently, in Rho-0 mutants with lower membrane potential, therefore, it is important to investigate the actual ROS/H₂O₂ levels during uncoupler treatments, particularly in both Rho+ and Rho0 genotypes.

It is worth noting that the OXPHOS system is a major source of ROS production and deletion of specific genes, such as cox4 (CIV), could enhance ROS levels due to electron leakage via CIII. However, in Rho0 mutants where a significant portion of the OXPHOS system is already deactivated, OXPHOS activity may decrease, leading to ETC blockage and subsequent lower membrane potential, which may lead to reduced ROS production in Rho-0 cells. Given this scenario, it would be important to address the concern of ROS levels in Rho0 mutants lacking mitochondrial DNA. To justify the conclusion "Altered ROS is not the signal for the G1-to-S phase progression delay in p0 cells," I recommend some additional dye-based methods to measure actual ROS/H₂O₂ levels, such as H₂DCFDA/Amplex Red, and more specific probes like mito-SOX.

Next, using antioxidants, authors showed a transient accumulation of cells in G1 phase. interestingly, the exogenous addition of antioxidants leads to a decrease in mitochondrial membrane potential in rho+ cells, similar to the effect observed with the uncoupler CCCP. However, I would like to raise a concern regarding the lack of direct experimental evidence supporting the role of antioxidants in reducing ROS levels. Taken together, the possibility of ROS involvement cannot be completely eliminated in this scenario. To address this concern, I request that the authors provide additional experimental data to further support their claim. Furthermore, it is worth noting that the impact of oligomycin has been limited to the Rho+ strain only, and no results has been shown on the mutant (rho0) strain. Considering that most of the mitochondrially encoded genes are already deactivated in the mutant strain, it is possible that oligomycin may have no further effect on membrane potential or ROS production. Please comment! It is also important to address that oligomycin could potentially enhance cellular ROS levels (Yi, Dae-Gwan et al., 2018; <https://doi.org/10.1371/journal.pone.0198619>).

I kindly request the authors to comment on the involvement of the RTG-dependent mitochondrial signalling in the G1/S transition during cell cycle progression.

What is the difference between "Relative $\Delta\Psi_m$ ($\Delta TMRE/NAO$) and Relative TMRE intensity" please explain and check their scales in the figures.

Reviewer #3 (Comments to the Authors (Required)):

It was known that the cell cycle is delayed in $\rho 0$ cells, but what triggers this delay has long remained unclear. In this manuscript, Gorospe et al. clarified that the decrease in mitochondrial membrane potential is the determinant of the delay in the G1-to-S transition. In $\rho 0$ cells (or cells treated with an uncoupler CCCP), the G1-to-S transition is delayed, but this delay is rescued by a hyperactive mutation of ATP1-111, which restores the mitochondrial membrane potential. Gorospe et al. also showed that cellular oxidative stress, a possible candidate for affecting the cell cycle, is not directly related to the delay in the G1-to-S transition. These findings, indicated by simple, reliable and well-controlled experiments, will contribute to the research field of mito-cellular signaling. Therefore, this manuscript is considered suitable for publication in Life Science Alliance.

Specific comments are as follows:

Major comments

(1) Figures 4B and 4C are the most important data in this manuscript, but these experiments were performed under unsynchronized conditions. Therefore, it is difficult to describe that "the ATP1-111 mutation was able to rescue the cell cycle phenotype of the WT $\rho 0$ cell" in line 179, because "the cell cycle phenotype of the WT $\rho 0$ cell" means the delayed transition from G1 to S. The authors should investigate whether ATP1-111 rescues the G1-to-S transition. The same applies to Figures 4E and 4F. Related to this comment, this reviewer did not understand the relationship between Figures 4B/4C and Figure S5B. In Fig. S5B, the timing of the G1-to-S transition in ATP1-111 $\rho 0$ cells appears to be similar to that in WT $\rho 0$ cells.

(2) In Figure 5D, the relative $\Delta\psi_{\text{sim}}$ ($\Delta\text{TMRE}/\text{NAO}$) should be investigated as in Figures 3A and 4A. In addition, this reviewer wonders whether the TRME fluorescence itself is not affected by antioxidants, NAC and GSH. If possible, it is better to include control samples of +NAC+CCCP and +GSH+CCCP, which should be nearly equal to +CCCP.

(3) In Figure 6, if the authors describe the Whi5 dilution model, it is better to examine the WHI5 transcript.

Minor comments

(1) In line 163, "P" should be " ρ ".

(2) In lines 170 and 172, "1" of F1 should be subscripted.

(3) Y axis in Figure 2A, "in G1" is unnecessary.

Response to Reviewers' Comments

Life Science Alliance submission LSA-2020-02091

Mitochondrial Membrane Potential Acts as a Retrograde Signal to Regulate Cell Cycle Progression

We thank the Reviewers for their constructive comments and suggestions, as well as for the overall positive evaluation of our manuscript. We have carefully considered the comments and modified our manuscript accordingly, adding several new data elements. The changes are outlined in the below point-by-point list.

Reviewer #1

This manuscript is well-written about the relationship between mitochondrial membrane potential and cell cycle delay. Their ways to display experimental evidence are sound and solid for this. In brief, the authors reveal that mitochondrial membrane potential act as retrograde signals which affect the cell cycle progression of cells in the manuscript. They compared mtDNA-deficient rho0 cells with normal cells (rho+), and revealed that the cell cycle of yeast is delayed in G1 when mtDNA is deficient. They also observed how inhibiting or overexpressing various mitochondrial functions including ATP synthesis, oxidative stress regulation, and mitochondrial membrane potential), in which mitochondria are mainly involved to affect the cell cycle. Although, they didn't identify specific pathway to regulate cell cycle delay from mitochondrial membrane potential, this manuscript clearly notes the effect of mitochondrial membrane potential on the cell cycle. I recommend to accept the manuscript as it is.

We thank Reviewer #1 for the positive evaluation and their interest in our work. We agree that identifying the specific pathway mediating the regulation of the cell cycle in response to mitochondrial membrane potential loss reported here is an important avenue and work is ongoing to in the future be able to uncover this pathway.

Reviewer #2

The manuscript entitled "Mitochondrial membrane potential acts as a retrograde signal to regulate cell cycle progression" investigated how the mitochondria-related parameters affect cell cycle progression in wild-type Rho+ and Rho-0 mutant, lacking mitochondrial DNA in yeast. I have carefully examined the manuscript and I would like to commend the authors on their well-written and well-documented manuscript, which effectively utilizes genetic and pharmacological approaches to examine the role of mitochondrial membrane potential in the progression of the cell cycle. Overall, the experimental plans are sound, and the data presented is robust. However, it seems that some of the conclusions reached lack direct experimental evidence, despite their significance:

Using uncouplers, authors demonstrated that a $\Delta\Psi_m$ collapse causes G1-to-S phase transition delay, similar to the one observed in Rho0 cells and concluded the role of mitochondrial membrane potential in cell cycle progression. To ensure a thorough understanding of the underlying mechanisms, it is crucial to eliminate other mitochondria-related parameters for e.g., ROS and ATP in cell cycle progression. For instance, mild uncoupling has been shown to

typically lower ROS levels due to increased respiratory activity (DOI: 10.1074/jbc.M408918200), consequently, in Rho-0 mutants with lower membrane potential, therefore, it is important to investigate the actual ROS/H₂O₂ levels during uncoupler treatments, particularly in both Rho+ and Rho0 genotypes.

We thank Reviewer #2 for the positive assessment and the constructive criticism of the manuscript. We agree that uncoupler treatment will have wide-spanning effects on mitochondria and have now carried out the suggested direct measurements of mitochondrial superoxide using MitoSOX to address the effect of uncoupler treatment on mitochondrial ROS. This new data, presented in Fig. S4F, shows decreased MitoSOX signal in both ρ^+ and ρ^0 cells following uncoupler treatment (BAM15; green bars). However, the MitoSOX signal is comparable in WT ρ^+ and ρ^0 cells, as well as in ATP1-111 ρ^+ and ρ^0 cells (new Fig. 5F, also discussed in the next comment). Therefore, the MitoSOX signal, a measure of mitochondrial ROS, does not correlate with the G1-to-S delay (observed in BAM15 treated cells [with decreased MitoSOX signal] and in WT ρ^0 but not in any ρ^+ strain nor in ATP1-111 ρ^0 [all showing comparable MitoSOX signal]).

From Fig. S4F:

From Fig. 5F:

It is worth noting that the OXPHOS system is a major source of ROS production and deletion of specific genes, such as *cox4* (CIV), could enhance ROS levels due to electron leakage via CIII. However, in Rho0 mutants where a significant portion of the OXPHOS system is already deactivated, OXPHOS activity may decrease, leading to ETC blockage and subsequent lower membrane potential, which may lead to reduced ROS production in Rho-0 cells. Given this scenario, it would be important to address the concern of ROS levels in Rho0 mutants lacking mitochondrial DNA. To justify the conclusion "Altered ROS is not the signal for the G1-to-S phase progression delay in ρ^0 cells," I recommend some additional dye-based methods to measure actual ROS/H₂O₂ levels, such as H₂DCFDA/Amplex Red, and more specific probes like mito-SOX.

We now complement the indirect measure of oxidative stress (using lipid peroxidation as a readout) with direct measures of ROS as suggested by the reviewer. We have carried out both H₂DCFDA and MitoSOX measurements on our panel of strains, with the results shown below and in Figure 5E-F. The MitoSOX measurements of mitochondrial superoxide (new Fig. 5F) show no significant changes in mitochondrial superoxide production in ρ^+ or ρ^0 variants of our strains (WT, *ATP1-111* and *cox4Δ*), which is in good agreement with the original lipid peroxidation data (Fig. 5E). This data supports our original conclusion that the delayed cell cycle progression seen in wt ρ^0 and in *cox4Δ* ρ^0 , but not in *ATP1-111* ρ^0 , is not explained by alterations in ROS levels. On line 243 of the Results section, we now write: “Also levels of mitochondrial superoxide as indicated by MitoSOX fluorescence were comparable across our strains of interest (Fig. 5F; see Fig. S4F for controls).”

From Figure 5:

Even the H₂DCFDA measurements that are an indicator of many types of cellular ROS (H₂O₂, hydroxyl, peroxy and other ROS species) fail to show a correlation between ROS levels and timely cell cycle progression (Letter Figure 1). There is high DCF signal in *cox4Δ* ρ^+ that has normal G1-to-S progression, comparable in timing to that of WT ρ^+ . Furthermore, all three ρ^0 strains exhibit markedly increased DCF signal (Letter Figure 1) whilst the cell cycle defect is only observed in wt ρ^0 and in *cox4Δ* ρ^0 , not in *ATP1-111* ρ^0 . Thus, although the results of the H₂DCFDA-based measurements of cytosolic ROS are strikingly different from the above lipid peroxidation and MitoSOX data that measure different aspects of oxidative stress, the conclusion that levels of oxidative stress do not correlate with the timing of G1-to-S progression in these strains still stands.

Letter Figure 1. Total cellular ROS was probed using H₂DCFDA that detects many types of cellular ROS (H₂O₂, hydroxyl, peroxy and other ROS species). Early-log cells were harvested, washed twice with PBS and resuspended in PBS to an OD₆₀₀ of 0.5. Cells were incubated with 10 μg/ml H₂DCFDA at 30 °C for 1 h and analyzed by flow cytometry. Comparison between variants and strains was performed by one-way ANOVA. ****p<0.0001 compared to the corresponding ρ⁺; #####p<0.0001 or ns p>0.05 compared to the WT ρ⁺.

It is important to note that H₂DCFDA measurements are known to be prone to artefacts in situations that involve a change in cellular pH, as can be expected here when comparing ρ⁺ or ρ⁰ cells with very different mitochondrial energy metabolism (<https://doi.org/10.1016/j.redox.2018.101065>; <https://doi.org/10.1016/j.freeradbiomed.2006.05.006>). Therefore, we consider the lipid peroxidation and MitoSOX data more reliable in this experimental scenario that compares ρ⁺ and ρ⁰ cells, and do not include the H₂DCFDA data, that are known to be artefactual under these conditions, in the manuscript.

Next, using antioxidants, authors showed a transient accumulation of cells in G1 phase. interestingly, the exogenous addition of antioxidants leads to a decrease in mitochondrial membrane potential in rho+ cells, similar to the effect observed with the uncoupler CCCP. However, I would like to raise a concern regarding the lack of direct experimental evidence supporting the role of antioxidants in reducing ROS levels. Taken together, the possibility of ROS involvement cannot be completely eliminated in this scenario. To address this concern, I request that the authors provide additional experimental data to further support their claim.

As requested, we have carried out ROS measurement following treatment with the antioxidants NAC or glutathione, and observe a decrease in ROS levels in both cases (Letter Figure 2).

Letter Figure 2. Antioxidant treatment decreases ROS levels in WT ρ^+ cells. ROS levels following treatment with 30 mM NAC or 20 mM glutathione (GSH) for 30 min. Harvesting and H₂DCFDA staining were performed as in the legend to Letter Figure 1. Groups were compared using one-way ANOVA. ** $p < 0.01$; *** $p < 0.001$.

Furthermore, it is worth noting that the impact of oligomycin has been limited to the Rho+ strain only, and no results has been shown on the mutant (rho0) strain. Considering that most of the mitochondrially encoded genes are already deactivated in the mutant strain, it is possible that oligomycin may have no further effect on membrane potential or ROS production. Please comment!

We now include the data for oligomycin treatment of the WT ρ^0 strain in Figure S2B. Because oligomycin targets the Fo subunit that is missing in ρ^0 cells, it is not expected to have any effect on these cells. Accordingly, oligomycin treatment has no effect on the cell cycle profile of ρ^0 cells (Fig S2B). On row 114, we now write: "As expected based on the lack of a functional Fo subunit, ρ^0 cells were not impacted by oligomycin treatment (Fig. S2B)."

From Fig. S2B:

It is also important to address that oligomycin could potentially enhance cellular ROS levels (Yi, Dae-Gwan et al., 2018; <https://doi.org/10.1371/journal.pone.0198619>).

Oligomycin treatment slightly increased mt ROS levels in ρ^+ but not in ρ^0 cells. This new experimental data is now included in Figure S4F (shown in first comment to Reviewer 2).

I kindly request the authors to comment on the involvement of the RTG-dependent mitochondrial signalling in the G1/S transition during cell cycle progression.

As a major mito-cellular signaling pathway, the RTG pathway is the foremost candidate for regulating cell cycle progression in response to the loss of mitochondrial membrane potential. However, deletion of either *RTG2* or *RTG3*, both required for the function of the pathway, did not eliminate the cell cycle delay observed in ρ^0 cells, indicating that the RTG pathway is not required for the G1/S transition delay (Letter Figure 3). We have also ruled out the involvement of many other candidate genes in this process, and are now carrying out a genome-wide screen which we hope will in the future shed light on the identity of involved protein factors. Given that we so far have only negative results on the mediating proteins/pathways, they were not included in the current manuscript.

Letter Figure 3. The G1-S transition delay in ρ^0 cells does not require the RTG retrograde pathway. Quantification of the cell cycle profiles of WT, *rtg2* Δ and *rtg3* Δ ρ^+ and ρ^0 cells. Values represent the average of at least 4 independent experiments, and error bars indicate standard deviation. The G1 population in each ρ^+ and its ρ^0 variant was compared using the two-tailed Student's t-test. **** $p < 0.0001$, ** $p < 0.01$.

What is the difference between "Relative $\Delta\Psi_m$ ($\Delta TMRE/NAO$) and Relative TMRE intensity" please explain and check their scales in the figures.

The "Relative $\Delta\Psi_m$ ($\Delta TMRE/NAO$)" in Fig. 3A and Fig. 4A is TMRE signal normalized to mitochondrial mass (as measured by NAO). We use this measure when comparing across different yeast strains to correct for possible differences in mitochondrial mass. In contrast, "Relative TMRE intensity" as used in the old Fig. 5D was uncorrected for mitochondrial mass and used to present the TMRE signal following short pharmacological treatments in the WT strain where mitochondrial mass will not have time to be affected. For consistency and clarity, we now present even the data in Fig. 5D as "Relative $\Delta\Psi_m$ ($\Delta TMRE/NAO$)".

Reviewer #3

It was known that the cell cycle is delayed in ρ^0 cells, but what triggers this delay has long remained unclear. In this manuscript, Gorospe et al. clarified that the decrease in mitochondrial membrane potential is the determinant of the delay in the G1-to-S transition. In ρ^0 cells (or cells treated with an uncoupler CCCP), the G1-to-S transition is delayed, but this delay is rescued by a hyperactive mutation of ATP1-111, which restores the mitochondrial membrane potential. Gorospe et al. also showed that cellular oxidative stress, a possible candidate for affecting the cell cycle, is not directly related to the delay in the G1-to-S transition. These

findings, indicated by simple, reliable and well-controlled experiments, will contribute to the research field of mito-cellular signaling. Therefore, this manuscript is considered suitable for publication in Life Science Alliance.

We wish to thank the reviewer for the positive comments on the manuscript. We have addressed all comments as indicated below.

Specific comments are as follows:

Major comments

(1) Figures 4B and 4C are the most important data in this manuscript, but these experiments were performed under unsynchronized conditions. Therefore, it is difficult to describe that "the ATP1-111 mutation was able to rescue the cell cycle phenotype of the WT ρ^0 cell" in line 179, because "the cell cycle phenotype of the WT ρ^0 cell" means the delayed transition from G1 to S. The authors should investigate whether ATP1-111 rescues the G1-to-S transition. The same applies to Figures 4E and 4F. Related to this comment, this reviewer did not understand the relationship between Figures 4B/4C and Figure S5B. In Fig. S5B, the timing of the G1-to-S transition in ATP1-111 ρ^0 cells appears to be similar to that in WT ρ^0 cells.

In accordance with this excellent request of the Reviewer, we now include data on the G1-to-S transition of synchronized WT, ATP1-111 and *cox4Δ* ρ^+ and ρ^0 cells, which is presented in the new Fig. 4D (cell cycle histograms in Fig. S3E). These data show that the ATP1-111 mutation rescues the G1-to-S transition timing of ρ^0 cells to levels comparable to the *cox4Δ* ρ^+ cells.

From Fig. 4D:

On line 182, we now write: "Moreover, analysis of α -factor synchronized cells revealed that the ATP1-111 mutation was largely able to rescue the delay in G1-to-S transition observed in ρ^0 cells: while the percentage of G1 cells in cultures of WT and *cox4Δ* ρ^0 cells was still above 90% 30 min after release from synchrony, only 67% of ATP1-111 ρ^0 cells remained in G1 phase at the indicated timepoint (Fig. 4D; Fig. S3D)."

The data in the new Fig. 4D also uncovered a slight delay in G1-to-S transition in *cox4Δ* ρ^+ cells, which was not observed in the unsynchronized cell data in the original version of the manuscript. This reviewer suggestion and the new data has thus led to better resolution of differences between the strains, which we are very happy about and now comment on in the

main text on line 196 (new text underlined): “Accordingly, deletion of the *COX4* gene did not have a pronounced effect on the cell cycle profile of unsynchronized ρ^+ cells (Fig. 4B, C). However, *cox4 Δ* ρ^+ cells exhibited a somewhat delayed transition from G1-to-S in synchrony experiments (Fig. 4D; Fig. S3D) and a stronger, more sustained cell cycle response to BAM15 than WT or *ATP1-111* ρ^+ cells, indicating that a functional ETC aids in timely recovery from uncoupling (Fig. 4F, Fig. S3F).”

Finally, we understand the confusion regarding the relationship between Fig. 4B/4C and Fig. S5B. The data presented in Fig. S5B show the timing of the G1/S transition in a representative expression analysis experiment. The experiment is largely comparable to the data shown as the new Fig. 4D above with the exception of a larger culture volume and thus slower sample handling. Both show partial rescue of the G1-to-S transition in *ATP1-111* ρ^0 cells compared to WT ρ^0 cells, but the difference is more obvious in the new Fig. 4D that is based on multiple replicates. Furthermore, the data in the new Fig. 4D reaches over a longer time period, thus providing a more precise indication of cell cycle timing.

(2) In Figure 5D, the relative $\Delta\psi_{\text{sim}}$ ($\Delta\text{TMRE}/\text{NAO}$) should be investigated as in Figures 3A and 4A. In addition, this reviewer wonders whether the TMRE fluorescence itself is not affected by antioxidants, NAC and GSH. If possible, it is better to include control samples of +NAC+CCCP and +GSH+CCCP, which should be nearly equal to +CCCP.

As suggested, we now present the membrane potential in Fig. 5D as “Relative $\Delta\psi_{\text{m}}$ ($\Delta\text{TMRE}/\text{NAO}$)”. Moreover, we include new data showing the TMRE fluorescence after the suggested treatments as the new Fig. S4D. This new data shows that TMRE fluorescence following NAC+CCCP or GSH+CCCP is comparable to that of CCCP alone, ruling out the possibility of TMRE signal being affected by the antioxidants. While all three treatments below severely decrease TMRE intensity, there is no statistically significant difference between CCCP-treated cells and those treated with a combination of CCCP and either NAC or GSH.

From Fig. S4D:

(3) In Figure 6, if the authors describe the Whi5 dilution model, it is better to examine the WHI5 transcript.

Transcript levels for the *WHI5* transcript are now shown in Fig. S5C, where no differences are observed between strains.

From Fig. S5C:

Minor comments

- (1) In line 163, "P" should be " ρ ".
- (2) In lines 170 and 172, "1" of F1 should be subscripted.
- (3) Y axis in Figure 2A, "in G1" is unnecessary.

Thank you for pointing these errors out, all three changes have been made.

August 28, 2023

RE: Life Science Alliance Manuscript #LSA-2023-02091R

Dr. Paulina H Wanrooij
Umeå University
Medical Biochemistry and Biophysics
Linnaeus väg 8
Umeå, - 90187
Sweden

Dear Dr. Wanrooij,

Thank you for submitting your revised manuscript entitled "Mitochondrial membrane potential acts as a retrograde signal to regulate cell cycle progression". We would be happy to publish your paper in Life Science Alliance pending final revisions necessary to meet our formatting guidelines.

-please add scale bars to Fig. S3A and S5A

A. FINAL FILES:

B. MANUSCRIPT ORGANIZATION AND FORMATTING:

****It is Life Science Alliance policy that if requested, original data images must be made available to the editors. Failure to provide original images upon request will result in unavoidable delays in publication. Please ensure that you have access to all original**

data images prior to final submission.**

The license to publish form must be signed before your manuscript can be sent to production. A link to the electronic license to publish form will be sent to the corresponding author only. Please take a moment to check your funder requirements.

Sincerely,

Reviewer #2 (Comments to the Authors (Required)):

The revised manuscript offers improved information and data, effectively addressing the concerns raised by the reviewers in the previous submission. I accept that the revised manuscript now contains ample data to substantiate the authors' claim that mitochondrial membrane potential serves as a potential retrograde signal during cell cycle progression. Therefore, I recommend accepting the revised MS in its current form.

Reviewer #3 (Comments to the Authors (Required)):

I have reviewed the revised manuscript and the point-by-point response to my previous comments. All my concerns have been appropriately addressed. I recommend publication of this work in Life Science Alliance.

August 30, 2023

RE: Life Science Alliance Manuscript #LSA-2023-02091RR

Dr. Paulina H Wanrooij
Umeå University
Medical Biochemistry and Biophysics
Linnaeus väg 8
Umeå, - 90187
Sweden

Dear Dr. Wanrooij,

Thank you for submitting your Research Article entitled "Mitochondrial membrane potential acts as a retrograde signal to regulate cell cycle progression". It is a pleasure to let you know that your manuscript is now accepted for publication in Life Science Alliance. Congratulations on this interesting work.

DISTRIBUTION OF MATERIALS:

Again, congratulations on a very nice paper. I hope you found the review process to be constructive and are pleased with how the manuscript was handled editorially. We look forward to future exciting submissions from your lab.

Sincerely,
